# Responding to harvest failure: Understanding farmers coping strategies in the semi-arid Northern Ghana

**David Boansi** [1] *, **Victor Owusu** [1], **Enoch Kwame Tham-Agyekum** [1], **Camillus Abawiera Wongnaa** [1], **Joyceline Adom Frimpong** [1], **Kaderi Noagah Bukari** [2]

1 Department of Agricultural Economics, Agribusiness and Extension. Kwame Nkrumah University of Science and Technology, Kumasi, Ghana, 2 Department of Peace Studies, School for Development Studies, University of Cape Coast, Cape Coast Ghana

* boansidavid@rocketmail.com, david.boansi@knust.edu.gh

**Data Availability Statement:** "The data underlying the results presented in the study are available from (https://wascal-dataportal.org/wascal_searchportal2/)."

## Abstract

Farmers coping strategies against harvest failures have implication for future adaptation to such shocks. Previous studies on farmers' vulnerability and response to shocks have emphasized on adaptation, at the expense of their coping to such shocks. Using a survey data from 299 farm households in northern Ghana, this study has analyzed farmers' coping strategies against harvest failure, and the drivers of the choice and intensity of the coping strategies. The empirical results show that most of the households used liquidation of productive assets, reduction in consumption, borrowing from family and friends, diversification of livelihoods, and migration to cities for off-farm jobs as coping measures in response to harvest failure. The empirical results from a multivariate probit model indicate that the choice of coping strategies is influenced by farmers' access to radio, net value of livestock produced per man-equivalent (ME), experience of yield loss in the previous year, farmers' perception about the fertility status of their crop fields, access to credit, distance to market, farm-to-farmer extension, location of the respondent, cropland per ME, and access to off-farm income. Empirical results from a zero-truncated negative binomial regression model also indicate that the number of coping strategies adopted increases with the value of farm implements, access to radio, farmer-to-farmer extension and being located in the regional capital. It however decreases with the age of the household head, number of family members abroad, a positive perception about the fertility status of crop fields, access to government extension services, distance to market, and access to off-farm income. Limited access to credit, radio, and markets renders farmers more vulnerable and pushes them to adopt more costly coping strategies. In addition, an increase in income earned from secondary livestock products decreases incentive for farmers to adopt liquidation of productive assets as coping strategy after a harvest failure. Policy makers and stakeholders could make smallholder farmers less vulnerable to harvest failure by enhancing their access to radio, credit, off-farm income and market, promotion of farmer-to-farmer extension, implementing measure to improve the fertility of crop fields in the study area, and enhancing farmers' engagement in the production and selling of secondary livestock products

**Funding:** This study was funded by the German Federal Ministry of Education and Research (BMBF) through the West Africa Science Service Center for Climate Change and Adapted Land Use (WASCAL). The funders had no role in study design, data collection and analysis, decision to publish, or preparation of the manuscript.

**Competing interests:** The authors have declared that no competing interests exist

## 1. Introduction

Most livelihood and vulnerability studies on farming households indicate that crop and livestock production are becoming riskier in recent decades [1–5] due to low productivity of croplands [6], population pressures on land-use [7, 8], institutional and technological challenges [2, 9, 10] and more importantly, increasing incidences of climate-induced productivity losses [3, 11, 12]. In sub-Saharan Africa for instance, majority of the farm households are so vulnerable that they hardly meet the basic nutritional needs for a healthy life in both favorable and less favorable seasons [7, 11, 13]. The sources of this vulnerability may come from over-dependence of farmers on rain-fed agriculture, limited access to off-farm employment opportunities [14, 15], limited access to safety nets [16,17], and the narrow scope of feasible coping alternatives available to farmers after a bad harvest [12, 13, 18]. Besides, the major agricultural zones in SSA are characterized by hot and dry weather conditions [19], with rain-fed agriculture under low-input conditions being the dominant source of food production and primary livelihood for the populace in these zones [3]. The continuous reliance of the farmers on rain-fed agriculture and other climate-sensitive enterprises for sustenance make them vulnerable to seasonal droughts and dry spells overtime.

Evidence from most studies [3, 6, 20–23] contend that vulnerable farming households in many countries in sub-Saharan Africa experience harvest failure. A work by [20] reveal that smallholder farmers in West Africa are susceptible to about 10% to 40% risk of probable harvest failure. Projections by [21, 22] and impact assessments by [3, 6, 23] indicate that sub-Saharan Africa could be subjected to an onslaught of bad harvests (harvest failures). Harvest failure, may be caused by extreme climate events such as droughts or floods [24], human health problems [19], and/or pests and diseases [19, 25, 26] (. Harvest failure undermine livelihoods and destabilize food prices [24, 27], could result in income and consumption losses, and in most rural contexts, induce the depletion of productive assets to smooth consumption [28–30]. With agriculture being the mainstay of at least 60% of the work force in the region [1, 3], these projections and the accompanying risk of increased harvest failures could undermine poverty alleviation and food security agenda in sub-Saharan Africa.

After experiencing harvest failure, farmers often use different combinations of coping strategies to ensure survival. Coping strategies are defined in this study as the short-term and immediate responses of farmers to shocks and are aimed at managing the risks associated with such shocks [31, 32]. While coping mechanisms assist farmers against shocks, most of the studies conducted so far to assess farmers' vulnerability to shocks and their responses have focused more on farmers' adaptation strategies at the expense of coping strategies, the latter of which have implications for farmers adaptation [28, 30]. Adaptation strategies encompass the reactive, concurrent or anticipatory long-term measures (which include changes in behavior and practices) implemented to help reduce vulnerability to shocks [33, 34].

Most of the studies have analyzed farmers' adaptation strategies to climate and weather shocks and their determinants [35–38]. The few studies that assessed how farmers cope with harvest failures and other environmental stressors [29, 39, 40] documented only the strategies farmers used without analyzing the determinants of the choice and intensity of the coping strategies (except for [28] in Ethiopia, [30] in Kenya, [25] in Malawi, and [32] in Ghana). Notable among the exceptions is a recent study by [32] which analyzed both the coping and adaptation strategies women farmers use to respond to extreme climate events, and the factors that influenced them to use those strategies. Research efforts to identify farmers' coping strategies in response to harvest failure and the factors that influence the choice of the coping strategies could guide policy makers to draft and implement policy-relevant measures to minimize agricultural production risks to smallholder farmers in sub-Saharan Africa. This study seeks to answer the following research questions, *Q1*: What are farmers' experiences of harvest failure,

and what coping strategies do they use to respond to them? *Q2*: What are the determinants of the choice of coping strategies? *Q3*: What factors influence the intensity of the strategies used by the farmers to cope with harvest failure?

This study contributes to expanding the literature on farmer responses to production related shocks in two ways. Firstly, in addition to identifying and documenting the strategies used by farmers in coping with production losses, this study analyzes the determinants of the strategies used. Secondly, the study identifies and documents the drivers of the intensity (number of strategies) of adoption of the coping measures. The rest of the paper are organized as follows; A review of how farmers cope with harvest failure in SSA and other developing regions of the world is provided in section 2. The materials and methods are discussed in Section 3. The results of the study are presented and discussed in Section 4. Conclusions and policy recommendations are provided in Section 5.

## 2. Literature review

The impact of harvest failures on food security and livelihoods are expected to worsen in countries and locations that are already suffering from high levels of hunger and poverty [41]. For most farm households in SSA and other developing regions of the world, the principal goal of coping with harvest failure is to maintain food security and income [42, 43]. Farmers' response to harvest failure generally involves adjustments in a specific action (e.g. migration, distress liquidation of productive assets, etc.), systematic change (e.g. on- and off-farm livelihood diversification, and changes in cropping pattern), institutional reform (e.g. provision of social safety nets), or processes (e.g. Learning about risks, evaluation of response, or creating an enabling condition) [44, 45]. Among the highly documented strategies used by farmers in coping with harvest failures in Africa and Asia are a reduction in consumption, the sale of productive assets, borrowing of cash and in-kind items from family and friends, migration, withdrawal of children from school to help with work on farm, and livelihood diversification [29, 30, 39, 40, 46–48].

The use of such strategies is induced by farmers' asset base, their dependence on small/medium-scale subsistence farming, and their limited access to off-farm employment opportunities, formal/informal credit, savings, insurance and/or other relevant safety nets [29, 49]. The consequent impacts of most of such strategies are long-lasting wealth shocks that push the vulnerable farm households and rural inhabitants into a poverty trap [46]. For example, while the liquidation of a fraction of productive assets such as livestock, land or tools could help to bridge present consumption gaps, the use of this strategy as a coping mechanism could compromise the future growth potential of farmers and worsen food production situation in subsequent years [50]. Most of these productive assets could take a very long time for them to be replenished and are generally sold at low prices due to excess supply in years of harvest failure [51]. A reduction in the current number, quality and quantity of meals and an increase in the number of children withdrawn from school to work on farm due to bad harvests could affect future human capital development, especially of young children in the household [48, 52]. While migration is generally perceived to be a necessary action after harvest failure (or bad harvest), it is mostly deemed an undesirable strategy for coping with shocks due to its ability to reduce the potential labor capacity of the households and over burden the members left behind with farm work, especially, women [48, 53]. This could affect future adaptation to climatic, economic and health shocks.

## 3. Methods

### 3.1. Conceptual framework

Risk has become an inherent attribute and a normal element in farming, especially in the West African region [54]. With increasing risk of climatic and production related shocks, has arose

a need for farmers to implement diverse adverse impact minimizing strategies ex-ante and/or ex-post. While emphasis has so far been placed on the adaptation strategies used by farmers, limited emphasis has been placed on how farmers cope with shocks, especially those related to agricultural production. This study sources to bridge this gap, by analyzing farmers experience of harvest failure in northern Ghana, their coping strategies and the determinants of the choice and intensity of the strategies used.

Farmers experience of harvest failure has been attributed in literature mostly to climatic shocks such as droughts or floods [24], pest and diseases [19, 25, 26], and to human health problems [19], the latter of which has implications for labour supply. When exposed to the risk of harvest failure, farmers adopt several coping strategies to smooth income, consumption or both. [55] outlined five main mechanisms that are sequentially adopted when households face shocks that threaten their food security. The first of such mechanisms is anticipatory in nature, and involves income diversification. The second draws on social networks. In instances where the first two mechanisms are found to be insufficient coping mechanisms, some household members migrate temporarily, while in households where migration becomes less of an option, a reduction in households food consumption is adopted. Besides migration and/or reduction in consumption, households may deplete agricultural assets such as implements and livestock. However, if all these mechanisms fail, most households tend to deplete fixed assets such as land or buildings.

In effort to inform policy decisions on the drivers of the choice of coping strategies employed by farmers after the experience of a bad harvest, and as shown in Fig 1, several variables have been found to be key determinants, among which are household and personal characteristics, past experience of harvest failure, institutional/infrastructural variables, farm characteristics, access to information, and geographic/locational variables. Among such variables are the sex, age and education of the household head, family size and distribution,

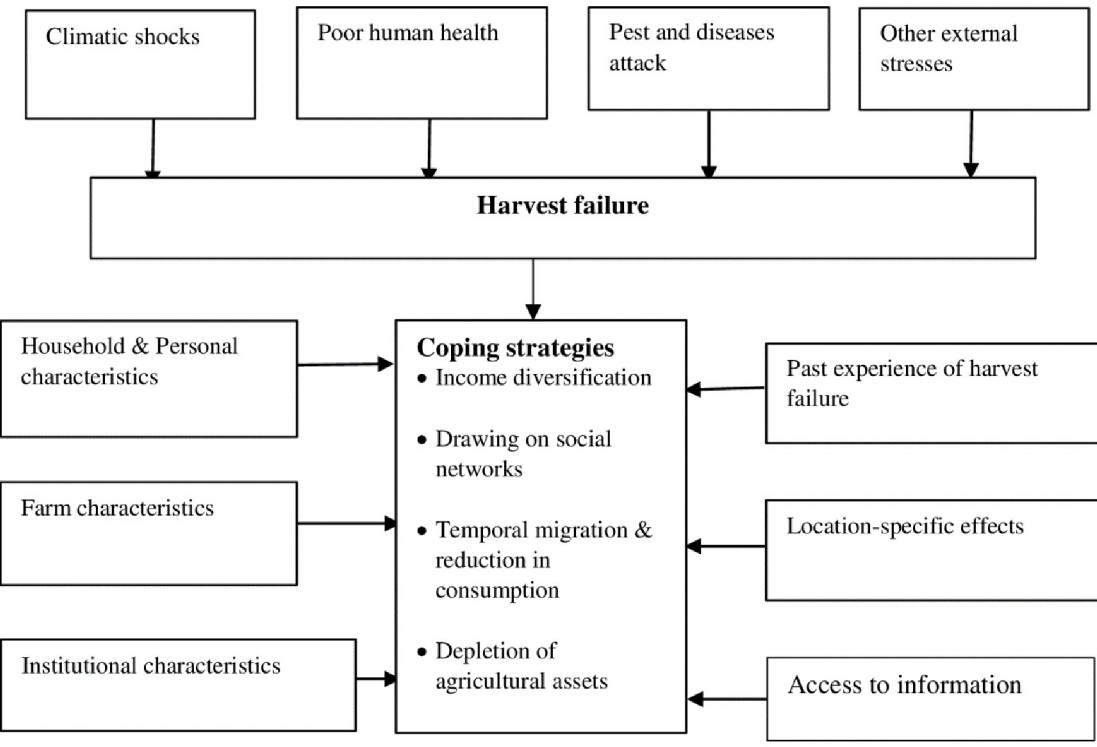

**Fig 1. Conceptual framework.** Source: Authors' construct.

livestock ownership, access to formal/informal credit and extension, access to off-farm employment/income, ownership of radio, social capital, access to markets, landholding, farmer-to-farmer extension, access to information on climate change, weather, agroecological zone, seniority within the community, perception on fertility of crop fields, access to remittances, farm income, and production orientation (subsistence or commercial) [28, 32, 48, 56].

The adoption of a reduction in food consumption as a coping strategy is found to be a common practice among women, smallholder farmers, households with literate and/or aged heads, and farmers with access to government extension services [57, 58]. Farmer-to farmer extension is found to enhance the selling of livestock and other productive assets to smooth consumption [28, 48, 56]. Besides this, the selling of productive assets as a coping mechanism has been found to be positively associated with the degree of dependence of the farm households on farming for their livelihood. It is reported that more diversified households are likely to be less reliant on selling livestock, farm implement, and land due to the availability of other coping mechanisms within their feasible set [18, 28].

The depletion of productive assets in years of adversity is as well reported to be a common practice among semi-commercial/commercial producers who produce to meet both consumption and cash needs of their households [18]. Under adverse weather conditions, this group of producers are subjected to crop yield losses and to potential income losses from market sales, placing on them a greater need to cope. The selling of valuable productive assets as a coping mechanism after a bad harvest is reported to be a common practice among larger households and households with relatively older heads [18, 28]. The latter association has been linked with the high reputational cost of borrowing or begging by the aged heads who mostly play seniority roles within the wider community [18] and to reduced mobility by these heads which rules them out of coping strategies like migration and livelihood diversification. In contrast to expectation, a study by [56] report that improved access to markets increases the likelihood of selling livestock. While the limited access of farmers to markets is logically associated with limited access to off/non-farm employment opportunities, thereby making the liquidation of productive assets a more likely coping mechanism in remote areas, the positive association found between improved market access and asset liquidation by [56] was attributed to the enhanced opportunity for farmers to purchase and sell goods in the markets.

Temporal/permanent migration by some members of the households to cities is found to be a common practice among households who have limited access to off-farm employment opportunities (Ashraf *et al.*, 2014), but it is less common practice among households with higher landholding per capita [29, 59]. While farmers with better access to credit are found to be less likely to migrate, those closer to local markets are found to be more prone to migration and engage in nonfarm activities as coping mechanisms after a harvest failure [59]. Farmers with stronger physical capital (livestock, farm implement, etc.) are found to be less likely to migrate [59]. Regarding the determinants of other coping mechanisms, [28] found that borrowing from relatives as a coping strategy is enhanced by farmers' access to government extension services and farmer-to-farmer extension. Besides this, they report an inverse association between farmers' access to credit and the seeking of off-farm opportunities (livelihood diversification) as a coping strategy. Ownership of radio is found to increase the probability of depleting assets and borrowing from relatives to cope with drought-induced harvest failure [28].

Studies that have analyzed farmers coping strategies after a shock have reported that the strategies adopted are mostly path-dependent [28, 30, 38], in that the adoption of some strategies may be dependent on the adoption of others. Some of the strategies are adopted to complement others, while others serve as substitutes. This implies a potential for farmers to adopt more than one strategy, necessitating the analysis of the determinants of the choice of coping strategies adopted, and the intensity of the strategies used.

### 3.2. Analytical framework

After the experience of harvest failure, the affected households try to implement mechanisms to cope with the impacts. Depending on the financial capacity or the level of vulnerability of the affected people, a variety of strategies may be employed to ensure the survival of the members of the household, and mostly at the expense of future investment capacity. This logically implies that the coping mechanisms implemented by farmers after the experience of a harvest failure may not be mutually exclusive, but the total number adopted will be. An understanding of the broad range of strategies implemented by farmers and the determinants of the choice and intensity of the strategies adopted could help to gain a deeper and clearer insight into farmers' vulnerability to harvest failure. Findings from this study could guide the proposition of measures to help reduce farmers' vulnerability to harvest failure in the study area and other vulnerable regions in SSA

Based on the results from this study, relevant policy recommendations are made, the implementation of which could help to reduce farmers' vulnerability to harvest failure. Based on an a priori assumption that farmers in the study area implement a variety of strategies to cope with harvest failure, and that the strategies are interdependent, a multivariate probit model is used for the analysis [60]. The model is expressed in its simplest form as follows:

$$C_{ik}^* = \propto_k Z_{ik} + \varepsilon_{ik}, \quad k = 1, \ldots \ldots, K \tag{1}$$

$$C_{ik} = \begin{cases} 1 & if\ C_{ik}^* > 0 \\ 0 & otherwise \end{cases}$$

where, from Eq (1), $C_{ik}$ is the use of coping strategy $k$ by household $i$, $C_{ik}^*$ is the latent propensity for household $i$ to employ coping strategy $k$, $Z_{ik}$ is a vector of variables presumed to affect the adoption of the respective coping strategies. The index $\propto_k$ represents intercept for the $k^{th}$ strategy, while $\varepsilon_{ik}$ are error terms that are multivariate normally distributed, with zero means, unitary variance and $n \times n$ correlation matrix (Mulwa *et al.*, 2017). Positive correlations among the error terms implies that the strategies involved are implemented as complements, while negative correlations indicate that they are adopted as substitutes. Based on evidences in literature [including 18, 28, 32, 37, 48], experience from the field, and expert opinions, a total of 22 variables (including a district dummy) are considered for the index $Z_{ik}$ (details are provided in subsequent sections). Although all the strategies implemented by the farmers are initially considered in the analysis, only the major coping strategies (implemented by at least 15% of the households) are considered in the multivariate probit model.

For a clearer insight into how farmers cope after a harvest failure, we also analyze the determinants of the intensity of farmers coping strategies. In this paper, we analyze the intensity of coping strategies as the number of coping strategies adopted by the farmers [36, 61, 62]. To estimate models with count data as dependent variable, several approaches, including Poisson regression, and truncated regression models have been used. In this study, a zero-truncated negative binomial regression is used due to the non-observance of zero counts in the number of strategies reported and the ability of the approach to model overdispersion in data, an attribute known to be very common in many count data [63–65]. The latter statement implies that while regression models based on a Poisson distribution assume equidispersion, (equality of mean and variance of the outcome variable), most count data usually have variance that are larger than the mean, and this is better corrected for by regression models based on negative binomial distribution and its generalizations [64].

In considering $Y_i$, $i = 1,2,\ldots, n$ to be a count variable that follows a discrete probability function $Pr(Y_i = y_i)$, and assuming $k + 1$ are omitted, a left-truncated distribution is observed and the probability function for such a distribution is given by [65]

$$f_k(Y_i) = Pr(Y_i = y_i | Y_i > k) = \frac{Pr(Y_i = y_i)}{Pr(Y_i > k)} = \frac{f(y_i)}{1 - F(k)}, \tag{2}$$

From Eq (2), $f_k(Y_i)$ is a representation of the truncated (above k) probability function, $f(y_i) = Pr(Y_i = y_i)$ represents the probability function of $Y_i$ (a random variable), and $F(k)$ represents the distribution function evaluated at $k$. Zero- or left-truncation ($k = 0$) is the most common form of truncation in count models, and for zero-truncation, Grogger and Carson (1991) proposed the following Poisson distribution:

$$Pr(Y_i = y_i | Y_i > 0) = \frac{e^{-\lambda_i}\lambda_i^{y_i}}{y_i!\left(1 - F_p(0)\right)} = \frac{e^{-\lambda_i}\lambda_i^{y_i}}{y_i!(1 - e^{-\lambda_i})} = \frac{\lambda_i^{y_i}}{y_i!(e^{\lambda_i} - 1)} \tag{3}$$

From Eq (3), the value of $Y_i = 0$ is omitted, $\lambda_i = \exp(x_i'\beta)$, $i = 1, 2, \ldots, n$ represents the mean of the Poisson distribution, $\beta$ is a $(k \times 1)$ vector of parameters, the index $x_i$ is a $(1 \times k)$ vector of covariates, and $F_p(0) = (1 - e^{-\lambda_i})$ represents the distribution function at 0. With this, the condition mean and variance of $Y_i$ are respectively as follows:

$$E(Y_i | X_i, Y_i > 0) = \lambda_i\left[1 - F_p(0)\right]^{-1}, = \lambda_i\left[1 - e^{-\lambda_i}\right]^{-1} \tag{4}$$

$$Var(Y_i | X_i, Y_i > 0) = E(Y_i | X_i, Y_i > 0) \times \left[1 - F_p(0)E(Y_i | X_i, Y_i > 0)\right], = \frac{\lambda_i}{1 - e^{-\lambda_i}}\left[1 - \frac{\lambda_i e^{-\lambda_i}}{1 - e^{-\lambda_i}}\right] \tag{5}$$

In over dispersed data, the estimates of the regression parameters for the truncated Poisson model will be biased and inconsistent [66], and to handle such overdispersion, a mixture model with over dispersed distribution like the negative binomial regression model which is a gamma Poisson mixture model is deemed appropriate and the distribution of the zero-truncated negative binomial model is given as follows [64, 65]

$$Pr(Y_i = y_i | Y_i > 0) = \frac{\Gamma\left(y_i + \frac{1}{\alpha}\right)}{\Gamma(y_i + 1)\Gamma\left(\frac{1}{\alpha}\right)}(\propto \lambda_i)^{y_i} \times (1 + \propto \lambda_i)^{-\left(y_i + \frac{1}{\alpha}\right)}\left[1 - F_{NB}(0)\right]^{-1}, \tag{6}$$

$$= \frac{\Gamma\left(y_i + \frac{1}{\alpha}\right)}{\Gamma(y_i + 1)\Gamma\left(\frac{1}{\alpha}\right)\left[1 - (1 + \propto \lambda_i)^{-\frac{1}{\alpha}}\right]}(\propto \lambda_i)^{y_i} \times (1 + \propto \lambda_i)^{-\left(y_i + \frac{1}{\alpha}\right)} \tag{7}$$

From Eqs (6) and (7), $F_{NB}(0) = (1 + \propto \lambda_i)^{-\frac{1}{\propto}}$, $\propto > 0$ represents the distribution function at 0. For the above distribution, the condition mean and variance of $Y_i$ are specified as follows:

$$E(Y_i|X_i, Y_i > 0) = \lambda_i[1 - F_{NB}(0)]^{-1}, = \lambda_i\left[1 - (1 + \propto \lambda_i)^{-\frac{1}{\propto}}\right]^{-1} \tag{8}$$

$$Var(Y_i|X_i, Y_i > 0) = \frac{E(Y_i|X_i, Y_i > 0)\left[1 - [F_{NB}(0)]^{1+\propto}E(Y_i|X_i, Y_i > 0)\right]}{[F_{NB}(0)]^{\propto}},$$

$$= \frac{\lambda_i}{1 - (1 + \propto \lambda_i)^{-\frac{1}{\propto}}}\left[1 + \propto \lambda_i - \frac{\lambda_i(1 + \propto \lambda_i)^{-\frac{1}{\propto}}}{1 - (1 + \propto \lambda_i)^{-\frac{1}{\propto}}}\right]^{-1} \tag{9}$$

## 3.3 Study area

Livelihoods in the Upper East region of Ghana are highly dependent on seasonal climatic conditions, especially rainfall. Located in the north-eastern corner of Ghana, the Upper East region is well noted to produce traditional staples like sorghum and millet, and is a major player in the country's production and supply of maize, rice, and groundnuts. The region also serves as a fertile ground for the production of cattle, sheep, goats, chicken, guinea fowl, pigs and donkeys. Agriculture is the major source of employment for about 84% of the region's total population [67]. Farmers in the region primarily earn agricultural income from the production and sales of crops and secondary livestock products like eggs, draught services, milk, wool and leather. Occupying a total land area of 8,842 km$^2$, and a population density of 120 people/km$^2$ [68], the Upper East region is characterized by a unimodal rainfall regime, with a growing period between May and October, and with maximum monthly rainfall accumulations in August and September [54].

As shown in Fig 2, the total seasonal rainfall for the region ranges between 703 mm (in the year 2013) and 1,073 mm (in the year 1999), with a mean estimate of 905.8 mm and a coefficient of variation of 10.6%.

## 3.4 Data and sampling

The study uses data from a household survey conducted by the corresponding author to assess farmers' experiences of harvest failure, their coping strategies and the determinants of the strategies adopted. A total of 22 communities (see Table A1 in the S1 Appendix for details) across 5 districts in the Upper East region of Ghana are covered in this study. Based on a multi-stage sampling technique, the survey commenced with a random sampling of 5 districts out of 13 (as of 2014/2015). The selected districts are the Kassena-Nankana East, Kassena-Nankana West, Bolgatanga Municipal, Talensi district and the Nabdam district. From the district level and based on expert opinions on the level of agricultural production in each of the selected districts, population density and proneness of the districts to crop failures, a total of 6 communities were randomly selected from the list for the Kassena-Nankana East, and 4 each for the remaining districts. Based on a farmer/farm household list provided by the district MoFA offices for the selected communities, based on an enumeration exercise that preceded the survey, a total of 70 farmers/farm households were selected for the Kassena-Nankana East, 60 for the Kassena-Nankana West (*with one dropped from the current study due to detected oddities in the response variables and other independent variables*), and 90 for the Bolgatanga Municipal. For the Talensi and Nabdam districts where authorities could not provide a list of farmers as of the time of the survey, 40 farm households were systematically selected in each of the districts starting with the first house and every other third thereafter. Respondents were

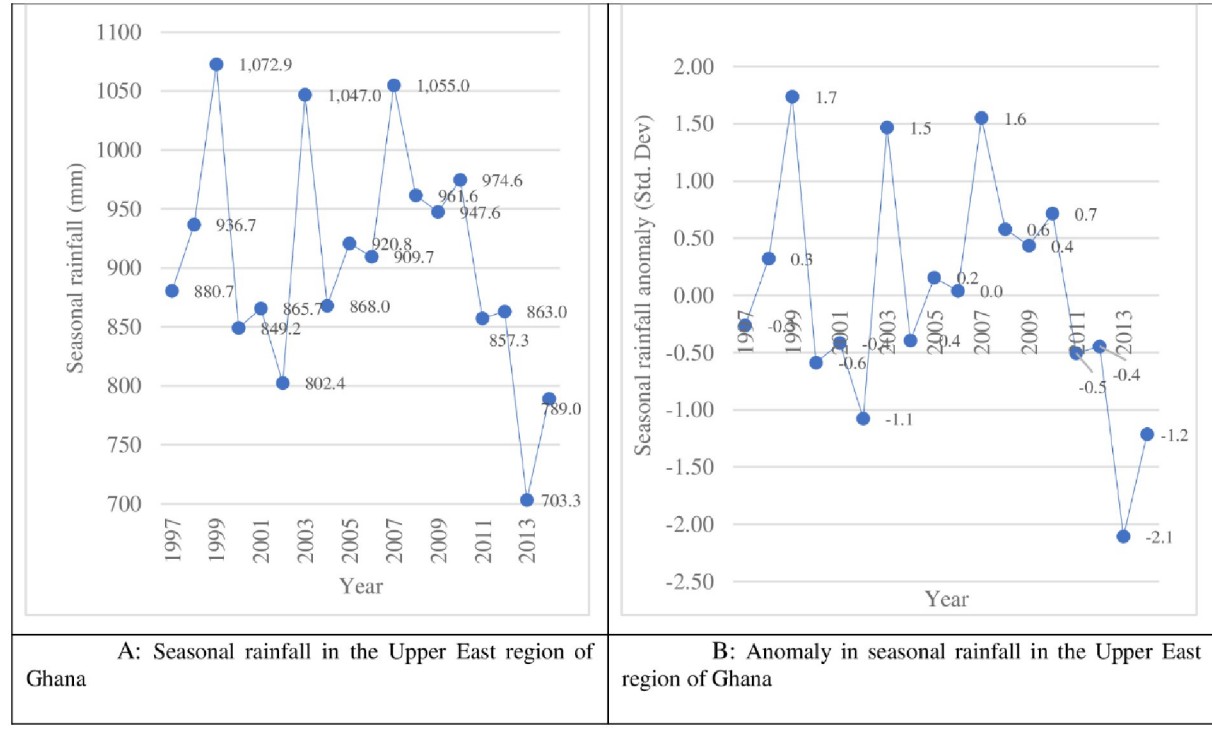

A: Seasonal rainfall in the Upper East region of Ghana

B: Anomaly in seasonal rainfall in the Upper East region of Ghana

**Fig 2. Rainfall in the Upper East Region of Ghana.** Source: Authors (with data from NASA's climatological database).

interviewed by trained research assistants (most of which were Extension Officers). The main survey was preceded by a pre-test with 30 farm households.

During the survey, emphasis was placed on gathering data on household demographics, socio-economic and farm-level variables, farmers' response on their cropping and livestock activities, their experiences of harvest failure after the less favorable agricultural season of the year 2014, previous experience of bad harvest, observed changes in crop yields relative to the norm, and farmers' involuntary responses (coping strategies) after the experience of harvest failure. Data on institutional and infrastructural variables like farmers' access to government extension services, formal/informal credit and markets were also gathered. Although farmers reportedly coped with harvest failure in different years, emphasis is placed in this study on their response to harvest failure in the year 2014, correcting however for the potential effect of a similar experience in the preceding year. The data for the study was collected between February and May 2015, with the gathering preceded by a pre-test between November and December 2014. Crops are usually harvested between October and December each year, and the time for the data collection for this study was at least 2 months after the harvest and general agricultural production for the year. In gathering data on farmers experience of harvest failure, emphasis was placed on experiences between the years 2005 to 2014.

The data for this study was collected as part of a big dataset gathered by the corresponding author for his PhD work between the years 2013 to 2019. The work was first approved by the Research Ethics Committee for the Center for Development Research (ZEF, Bonn, Germany) in 2014 before the data collection began to ensure adherence to all approved ethics considered in research involving human participants. In addition to this, respondents' participation in the interviews conducted for this data was voluntary and none was forced to provide confidential information. Besides the implementation of measures to keep the respondent's identity

confidential, all interactions between the interviewers and interviewees, as well as sensitive information were kept confidential. Respondents gave both written (via signature/thumbprint) and oral consents before each data collection began.

## 4. Results and discussion

In this section, some descriptive statistics are presented on the farm households. In addition, we present and discuss results for the assessment of farmers' experiences of harvest failure, the perceived impacts, the coping strategies adopted, and the determinants of the choice and intensity of the strategies employed.

### 4.1 Characteristics of the farm households

Households in the study area are mostly headed by males. As shown in Table 1, approximately 85% of the household heads are men, and the average head is around the age of 52 years, with 3 years of schooling. Approximately 77% of the farmers have access to off-farm income, but only 27% have access to formal/informal credit. A man in the average household earns about US$14 from the sales of secondary livestock products per year (eggs and draught services). About 32% of the farmers belong to a cooperative/agricultural union, 87% have access to extension services, and the average household is about 4.9 km from the nearest daily market place. Per the consumer/worker ratio in Table 1, a household in the study area has approximately 66% more mouths to feed than the number of hands available to work. Per the orientation of crop production in the region, about 74% of the farmers produce crops on a subsistence basis. About 66% of the farmers have access to radio, while 75% engage in farmer-to-farmer extension. The cropland per man-equivalent (ME) for the study area is estimated at 0.51 ha. After accounting for livestock mortality, the net value of livestock produced per ME for the study area is estimated at US$489.4, while the net crop income per ME is estimated at US$43.2. Of the 299 farm households covered in this study, about 76% perceive that they produce on fertile croplands.

### 4.2 Experience of harvest failure and farmers coping strategies

This section identifies and documents farmers' experiences of harvest failure, perceived impacts and coping strategies adopted by the farmers. From Fig 3, it is found that while all the farmers reportedly experienced a harvest failure in the year 2014, about 22.7% of the farm households also recorded a harvest failure in the year 2013. A harvest failure is conceptually defined in this study as the experience of a major reduction in the yield of the major staple crops compared to what is usually observed by the farmers. For the years 2013 and 2014, the farmers reported major decreases in the yields of maize, early millet, late millet, sorghum, rice and groundnut. The reported decreases in crop yields for the year 2013 ranged between 42.7% (for maize) and 61.5% (for early millet). For the year 2014 however, yield decreases ranged between 46.8% (for sorghum) and 56.8% (for rice).

After the experiences of harvest failure, and as shown in Fig 4, majority of the farm households adopted between 3 (25.75% of households) to 4 (60.87% of households) coping strategies to either smooth consumption, income or ensure the survival of the members. The major strategies adopted/implemented by the farmers were the sales of livestock and other non-land assets (94% of households), reduction in household's food consumption (86% of households), borrowing (food, cash, and other items) from family and friends (57% of households), livelihood diversification (54% of households), and migration of some household members to the city (15.7% of households). These observations are in line with documented evidences in

**Table 1. Descriptive statistics of variables used in the regression models.**

| Variables | Definition of variables | Mean | Std Dev |
|---|---|---|---|
| *Personal & household characteristics* | | | |
| Age | Age of the household head (Years) | 51.66 | 14.03 |
| Gender | Dummy = 1 if household is headed by a male, 0 otherwise | 0.846 | 0.361 |
| Education of household head | Years of schooling | 3.217 | 4.277 |
| Consumer/worker ratio | Index (ME of consumers /ME of workers) | 1.660 | 0.380 |
| Family members abroad (outside the community) | Count (potential sources of remittance) | 0.538 | 2.940 |
| Income from secondary livestock products per ME | US$/ME | 13.76 | 26.96 |
| Net value of livestock produced per ME | US$/ME | 489.4 | 633.6 |
| *Farm characteristics* | | | |
| Cropland per man-equivalent (ME) | Ha/ME | 0.507 | 0.397 |
| Ownership of land | Dummy = 1 if own land, 0 otherwise | 0.983 | 0.128 |
| Produce crops on a subsistence basis (Production of crops for household consumption) | Dummy = 1 if household produces crops on a subsistence basis, 0 otherwise | 0.742 | 0.438 |
| Value of farm implements (tools) | US$/household | 23.83 | 35.28 |
| Fertility of crop field | Dummy = 1 if fertile, 0 otherwise | 0.756 | 0.430 |
| Net crop income per ME | US$/ME | 43.23 | 189.5 |
| Experienced of severe yield loss also in 2013 | Dummy = 1 if household experienced harvest failure in 2013, 0 otherwise | 0.227 | 0.420 |
| *Institutional/infrastructural variables* | | | |
| Access to formal/informal credit | Dummy = 1 if household has access to credit, 0 otherwise | 0.271 | 0.445 |
| Access to government extension services | Dummy = 1 if household has access to government extension services, 0 otherwise | 0.873 | 0.334 |
| Distance to nearest market | Km from the nearest daily market | 4.945 | 3.618 |
| Access to off-farm income | Dummy = 1 if household has access to off-farm income, 0 otherwise | 0.773 | 0.420 |
| Group membership (Cooperative/agric. union) | Dummy = 1 if household head is a member of a cooperative/farmer union, 0 otherwise | 0.324 | 0.469 |
| *Access to information* | | | |
| Access to radio | Dummy = 1 if household has access to radio, 0 otherwise | 0.659 | 0.475 |
| Farmer-to-farmer extension | Dummy = 1 if household head partakes in a farmer-to-farmer extension, 0 otherwise | 0.753 | 0.432 |
| *District dummy* | | | |
| Bolgatanga (regional capital) | Dummy = 1 if household is located in Bolgatanga municipal, 0 otherwise | 0.301 | 0.459 |

NB: Exchange rate for the year 2014: 1 US$ = GHS 3.029

NB: For consumer/worker ratio, consumer is computed using the following conversion factors; for Females: 0-5years (0.40), 6–10 years (0.60), 11-17years (0.80), 18–65 years (0.80), > 65 years (0.70); for males 0-5years (0.40), 6–10 years (0.60), 11-17years (1.00), 18–65 years (1.00), > 65years (0.70); (modified version of age range proposed by Runge-Metzger and Diehl (1993)), while worker is computed as in Boansi *et al.*, (2017)

Source: Authors' computations (based on household survey data)

literature, including [47] for Burkina Faso, [28] for Ethiopia, [39] for Ghana, [40] for Zimbabwe, [29] for Madagascar, [30] for Kenya, and [48] for South Asia.

For a deeper insight into the adoption of various coping strategies by the respondents, a district level comparison was performed to ascertain potential heterogeneity in the choice and intensity of the strategies adopted. As shown in Table 2, it is found that except for the Nabdam district (*where about half of the respondents adopted 2 coping strategies*), most of the households in the remaining districts adopted 3 to 4 coping strategies. While strategies like the selling of livestock/assets, and reduction in consumption are common across all the 5 districts,

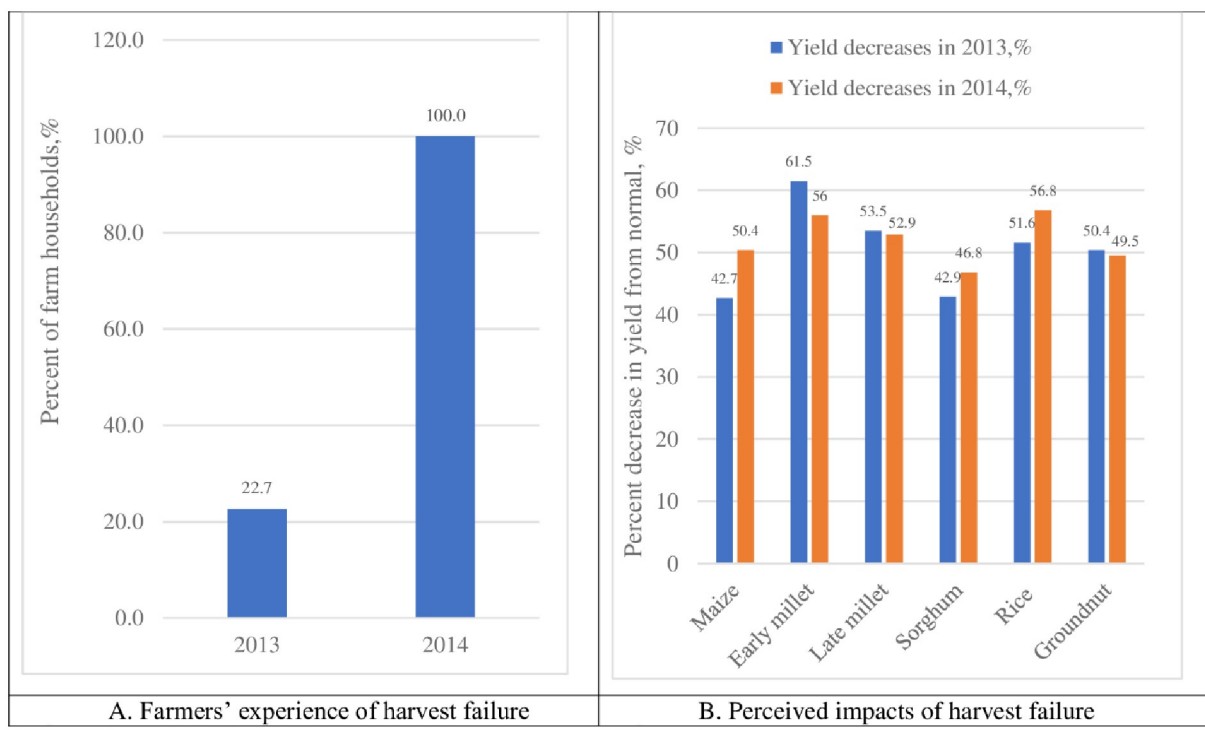

**Fig 3. Farmers' experience of harvest failure and perceived impacts.** Source: Authors' construct (based on household survey data).

borrowing from family and friends is a more common practice in the Bolgatanga Municipal (which happens to be the regional capital). Livelihood diversification is fairly adopted in all five districts, while the practice of laying off some laborers is only found in about a quarter of the households in the Kassena-Nankana East and West districts. Although the option for migration is found in less than 2% of the households in the Bolgatanga Municipal, about 28.8% and 50% of the households in Kassena-Nankana West and Talensi districts adopted this strategy to cope with harvest failure.

Although it may be assumed generally that the rational approach to cope with harvest failure would have been for households to first employ strategies that would not erode their productive assets (in line with the coping process outlined by [55]), it is found that the selling of livestock and other non-land assets is the main strategy used by the farmers in Upper East Ghana. This is in contrast with reports by [69–71], who argue that the depletion of productive assets is only employed as a coping strategy when households have limited portfolio of coping mechanisms. It is however in line with the findings of [56, 72, 73], who report that the selling of livestock is the main strategy households turn to in coping with production losses due to their use of livestock as a liquid savings. Besides, in the study area, livestock accounts for about 95.4% of the value of non-land assets held by the farmers (interviewed), making the depletion of productive assets a likely but undesirable option for coping with harvest failure. In addition, livestock generally represents wealth in northern Ghana and serves as an important insurance mechanism for consumption smoothing [39, 74]. The reported reductions in household food consumption by the farmers entailed a reduction in the quality, quantity and frequency of meals eaten in a day. In addition to migration (as an indirect form of livelihood diversification), 54% of the farmers reported that they made effort to diversify their income sources from sole-reliance on farming to other non-farm activities after the harvest failure. Beside these,

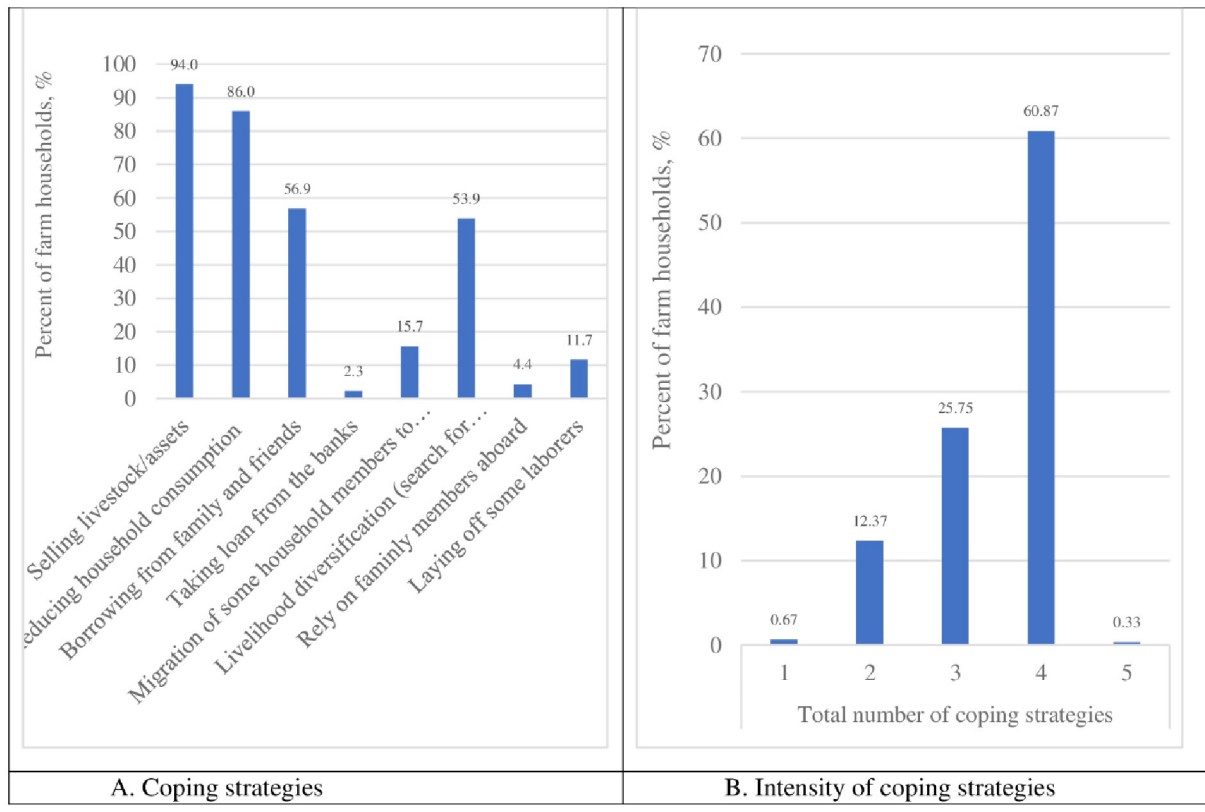

**Fig 4. Coping with harvest failure by farmers in Northern Ghana.** Source: Authors construct (based on household survey data).

some of the farmers (57%) depended on family and friends for loans, food and other items for survival. In line with the work of [18], it is found that majority of the farmers jointly used a variety of strategies to cope with harvest failure, and this necessitates the use of a model that accounts for the correlations between the strategies employed.

## 4.3 Determinants of farmers coping strategies

This section presents the results from the estimation of the multivariate probit model. This model is used to identify the determinants of the five main strategies farmers used to cope with harvest failure in the study area. The model helps to account for the potential correlations between the strategies, thereby minimizing/precluding potential biases in the estimation process.

Except for the years of schooling and membership in agricultural cooperatives/unions, all the other variables considered in the model had significant effect(s) on at least one of the dominant coping strategies considered in the model. Among the variables with significant effects however, access to radio, net value of livestock produced by ME, experience of yield loss in the previous year, farmers perception about the fertility status of their crop fields, access to credit, distance to market, farmer-to-farmer extension, the district (Bolgatanga) dummy, cropland per ME and access to off-farm income appeared to be the most important (with significant effects on at least two of the five dominant strategies). Farmers in the regional capital (Bolgatanga) were found to be more likely to borrow from family and friends to cope with harvest failure, but less likely to migrate in search of jobs in other cities and less likely to opt for a reduction in their food consumption.

**Table 2. District level comparison of choice and intensity of coping strategies.**

| Strategies | Choice | Kassen-NE (N = 70) | Kassen-NW (N = 59) | Bolgatanga Municipal (N = 90) | Talensi District (N = 40) | Nabdam District (N = 40) | Total (N = 299) |
|---|---|---|---|---|---|---|---|
| Number of coping strategies | 1 | 0.00 | 0.00 | 0.00 | 0.00 | 5.00 | 0.67 |
| | 2 | 15.71 | 1.69 | 4.44 | 2.50 | 50.0 | 12.37 |
| | 3 | 37.14 | 13.56 | 33.33 | 12.50 | 20.0 | 25.75 |
| | 4 | 47.14 | 84.75 | 62.22 | 82.50 | 25.0 | 60.87 |
| | 5 | 0.00 | 0.00 | 0.00 | 2.50 | 0.00 | 0.33 |
| Selling livestock /assets | No | 4.29 | 10.17 | 5.56 | 0.00 | 10.0 | 6.02 |
| | Yes | 95.71 | 89.83 | 94.44 | 100 | 90.0 | 93.98 |
| Reducing consumption | No | 11.43 | 6.78 | 21.11 | 0.00 | 27.5 | 14.05 |
| | Yes | 88.57 | 93.22 | 78.89 | 100 | 72.5 | 85.95 |
| Borrowing from family and friends | No | 75.71 | 47.46 | 3.33 | 50.0 | 62.5 | 43.14 |
| | Yes | 24.29 | 52.54 | 96.67 | 50.0 | 37.5 | 56.86 |
| Taking loan from the banks | No | 95.71 | 98.31 | 97.78 | 100.0 | 97.5 | 97.66 |
| | Yes | 4.29 | 1.69 | 2.22 | 0.00 | 2.50 | 2.34 |
| Migration of some household members | No | 91.43 | 71.19 | 98.89 | 50.0 | 92.5 | 84.28 |
| | Yes | 8.57 | 28.81 | 1.11 | 50.0 | 7.50 | 15.72 |
| Livelihood diversification | No | 34.29 | 40.68 | 57.78 | 37.50 | 57.50 | 46.15 |
| | Yes | 65.71 | 59.32 | 42.22 | 62.50 | 42.50 | 53.85 |
| Rely on family members abroad | No | 90.0 | 94.92 | 98.89 | 97.50 | 97.50 | 95.65 |
| | Yes | 10.0 | 5.08 | 1.11 | 2.50 | 2.50 | 4.35 |
| Laying off some laborers | No | 74.29 | 71.19 | 100.0 | 100.0 | 100.0 | 88.29 |
| | Yes | 25.71 | 28.81 | 0.00 | 0.00 | 0.00 | 11.71 |

Source: Authors (based on household survey data)

From the empirical results in Table 3, a negative association is found between the age of the household head and the adoption of a reduction in consumption and livelihood diversification as coping strategies. Given the relatively less mobile nature of the aged, most of such farmers may not be able to diversify their livelihoods by engaging in off-farm employment and other income generating activities. Due to the potential health implications of reducing consumption and the likelihood for the aged to have accumulated enough assets for consumption smoothing in times of adversity, the aged may not opt for a reduction in consumption after a harvest failure, but rather seek other alternatives. Households with male heads are found to be more likely to promote the migration of some members to cities in search of job after a harvest failure. In households where the heads have more mouths to feed than the number of hands available to work, the farmers are less likely to sell livestock and other assets to smooth income and consumption. As an alternative means of reducing liquidity constraint, the selling of secondary livestock products like eggs and draught services, reduces the need for farmers to engage in the distress liquidation of livestock and other production assets after a harvest failure. This implies that measures to promote farmers' engagement in the sale of secondary livestock products could reduce their vulnerability and the need for them to sell off productive asset as a coping strategy. Farmers with higher livestock holding, as reflected by the net value of livestock produced per ME, are found to be more likely to sell off their productive assets (livestock and other non-land assets) and reduce the quantity, frequency and quality of food consumed, but less likely to engage in livelihood diversification. Due to the ability for harvest failure to reduce the availability of feed for livestock and to potentially induce the death of the

**Table 3. Determinants of choice of farmers' coping strategies.**

| Variables | Selling liv. & Other assets (1) | Reduce food consumption (2) | Borrow from family & friends (3) | Migration (4) | Livelihood diversification (5) |
|---|---|---|---|---|---|
| Age | 0.0052 (0.0087) | -0.0189*** (0.0066) | -0.0012 (0.0080) | 0.0132 (0.0084) | -0.0162*** (0.0062) |
| Gender | 0.1163 (0.3700) | -0.1921 (0.3069) | 0.1010 (0.2527) | 0.5912** (0.2682) | -0.0252 (0.2534) |
| Education of household head | -0.0354 (0.0326) | -0.0420 (0.0304) | -0.0026 (0.0228) | -0.0153 (0.0288) | -0.0031 (0.0236) |
| Consumer/worker ratio | -0.6989** (0.2922) | 0.0285 (0.2560) | -0.0561 (0.2447) | 0.3130 (0.2750) | -0.1575 (0.2511) |
| Family members abroad | 0.1088 (0.1326) | 0.0510 (0.0634) | -0.0375 (0.0265) | -0.0763 (0.0526) | -0.2005* (0.1106) |
| Income from sec. liv. prod, /ME | -0.0202** (0.0080) | -0.0061 (0.0061) | -0.0051 (0.0038) | 0.0050 (0.0046) | 0.0003 (0.0052) |
| Net value of liv. Produced/ ME | 0.0023*** (0.0008) | 0.0007** (0.0003) | 0.0001 (0.0002) | -0.0001 (0.0002) | -0.0004** (0.0002) |
| Cropland / ME | -0.2170 (0.3638) | -0.0011** (0.0005) | 0.2824 (0.3083) | -0.6462** (0.2760) | 0.2816 (0.2450) |
| Ownership of land | 1.4197 (0.8721) | -2.0780 (2.4029) | -0.3029 (0.6904) | -1.6566** (0.6408) | -0.1616 (0.6652) |
| Produce crops on a subsistence basis | -0.0416 (0.3669) | 0.6468*** (0.2321) | -0.0580 (0.2041) | -0.3921 (0.2560) | -0.6212*** (0.2117) |
| Value of farm implements (tools) | 0.0151* (0.0083) | -0.0020 (0.0044) | 0.0002 (0.0027) | 0.0001 (0.0034) | 0.0138*** (0.0036) |
| Fertility of crop field | -0.1851 (0.2669) | 0.4466 (0.3002) | 0.4111* (0.2104) | -1.2046*** (0.2493) | -0.5764** (0.2437) |
| Net crop income/ME | -0.0002 (0.0008) | -0.0011** (0.0005) | -0.0007 (0.0005) | -0.0006 (0.0005) | 0.0017*** (0.0006) |
| Severe yield loss also in 2013 | 0.2031 (0.2889) | -0.2178 (0.2504) | -0.5091** (0.2094) | -0.5363** (0.2641) | 0.9100*** (0.2280) |
| Access to credit | -0.6557** (0.3061) | 0.5450* (0.327) | 0.1985 (0.2136) | -0.4653* (0.2524) | -0.0524 (0.2177) |
| Access to gov. extension services | 0.2484 (0.3704) | 0.435 (0.286) | -0.4640* (0.2481) | -0.3201 (0.2614) | -0.2504 (0.2789) |
| Distance to nearest market | 0.1183** (0.0539) | 0.0153 (0.0369) | -0.0318 (0.0210) | -0.0680* (0.0404) | -0.0527* (0.0305) |
| Access to off-farm income | -0.3860 (0.3938) | 0.4430* (0.2387) | -0.4860** (0.2447) | -0.2995 (0.2613) | -0.0752 (0.2144) |
| Group membership | 0.4218 (0.3141) | 0.0269 (0.2482) | -0.1336 (0.1879) | 0.3829 (0.2554) | 0.2539 (0.2002) |
| Access to radio | -0.9196*** (0.3217) | 0.6452*** (0.2334) | 0.6041*** (0.2037) | -0.7571*** (0.2467) | 0.1131 (0.1862) |
| Farmer-to-farmer extension | 0.1061 (0.2834) | 1.3038*** (0.2531) | 0.1062 (0.2214) | 0.7551*** (0.2772) | -0.1024 (0.2100) |
| Bolgatanga | 0.0627 (0.3069) | -0.4953* (0.2720) | 1.9973*** (0.3181) | -1.1294*** (0.3519) | -0.0976 (0.2118) |
| Constant | 1.0806 (1.4012) | 1.6811 (2.5521) | 0.2984 (1.0167) | 1.2320 (0.9479) | 2.3819** (0.9798) |
| *Statistics* | | | | | |
| Correlations (Rho) | (1) | (2) | (3) | (4) | (5) |
| (1) | 1.000 | | | | |
| (2) | -0.6028*** (0.1476) | 1.000 | | | |
| (3) | -0.0653 (0.1277) | -0.1307 (0.1462) | 1.000 | | |
| (4) | 0.0665 (0.1302) | 0.4317** (0.1698) | -0.5200*** (0.1131) | 1.0000 | |
| (5) | 0.1133 (0.1159) | -0.1173 (0.1223) | -0.2699*** (0.0948) | 0.1841 (0.1303) | 1.0000 |

Log pseudolikelihood = −468.11; Wald chi2 (110) = 778.1, Prob > chi2 = 0.000; ***, **, and * represent 1%, 5% and 10% significance level; Likelihood ratio test of rho21 = rho31 = rho41 = rho51 = rho32 = rho42 = rho52 = rho43 = rho53 = rho54 = 0:

Chi2 (10) = 38.37, Prob > chi2 = 0.0000

NB: ***, **, and * represent 1%, 5% and 10% significance level;

Source: Authors (based on output of multivariate probit model)

more sensitive herds/flocks, the disposal of live animals in times of adversity is not only a means to smooth income and consumption, but also an avenue for farmers to reduce carrying costs (Qureshi and Mujeeb, 2004). The observed association between livestock holding and the selling of productive assets as a coping strategy is in line with reports by [28, 56, 72, 73, 75]. While reports from India [including 46, 51show that in times of adversity, farmers use livestock inventory to reduce consumption shortfall, it is found in this study that the farmers with higher livestock holding opt for a reduction in consumption as a coping strategy. This may be attributed to the possibility that the quantity and value of livestock sold to smooth consumption may not be enough to bridge consumption gaps, or that the farmers choose to reduce their food consumption to prevent excess depletion of their valuable productive assets.

Households with higher cropland area per ME are less likely to encourage migration of some members in times of adversity, but more likely to reduce consumption of food. Having higher cropland area per ME is an indication of higher land-to-labor ratio and any decision to encourage migration of members will subject the household to further reduction in farm hands to work in subsequent seasons, which could have adverse implications for production. The effect of cropland per ME on migration is in line with reports by [29] and [59]. Land ownership is associated with a reduced likelihood for farmers to opt for migration as a coping strategy. Farmers who own land, and hence are relatively less vulnerable in terms of endowment could sell part of the land to smooth consumption and reduce the need for migration. Farmers who produce on a subsistence basis reduce food consumption after a harvest failure, relative to the semi-subsistence/commercial farmers, but are less likely to practice livelihood diversification. Compared to semi-subsistence or commercially oriented producers who experience both crop yield and income losses from market sales in times of adversity (thereby having a need to smooth both consumption and income), subsistence producers mostly suffer from only the former case (crop yield loss), and hence have a lesser need to diversify their livelihood. Having a higher value of farm implements and hence being more mechanized increases the chance for farmers to engage in the depletion of productive assets and to engage in livelihood diversification as coping strategies, the latter of which may be attributed to a potential for reallocation of household labor away from farm to off-farm work in times of adversity, and the possibility for machines to be used in the stead of household labor. Farmers with the perception of producing on fertile croplands are more likely to borrow from family and friends (with the hope of recovering and repaying soon), but are less likely to encourage the migration of some household members or diversify their livelihood. Farm households with relatively higher income per ME from crop production are more likely to engage in livelihood diversification after the experience of harvest failure, but less likely to opt for a reduction in food consumption. This implies that the need to reduce food consumption after a harvest failure is dependent on how much a farmer earned from his cropping activities for the year. Farmers who reportedly experienced harvest failure also in the year 2013 are found to be less likely to depend on borrowing and migration as coping strategies, but more likely to engage in livelihood diversification. The experience of harvest failure in two consecutive years could have led to a potential depletion of the assets of such households after the first experience, thereby reducing their feasible coping alternatives. In such a situation, diversifying one's livelihood by seeking employment in other income generating activities (in addition to farming) becomes a necessary coping mechanism.

Access to formal/informal credit reduces the possibility for farmers to opt for the sale of livestock and other non-land assets and migration as coping strategies due to a potential for credit to reduce liquidity constraint and to enable farmers to meet the cost of basic necessities in times of adversity. Having access to credit however promotes the adoption of a reduction in consumption as a coping strategy. While access to government extension services is associated with a reduced likelihood for farm households to borrow from family and friends, farmer-to-

farmer extension is associated with increased likelihood for the adoption of migration as a coping strategy, and with the likelihood for farmers to reduce food consumption [28]. A positive association is found between the selling of productive assets and distance to market, but a negative association between the distance variable and the adoption of migration and livelihood diversification. This is in line with an earlier report by [59]. Living farther from market centers limits farmers in their access to information on potential job opportunities, credit and other safety nets, which precludes them from opting for migration and livelihood diversification, and reduces their feasible coping set, thereby pushing farmers to engage in the sale of livestock and other assets to ensure survival, and smooth consumption and income. This implies that effort to improve farmers access to market could be a potential avenue to prevent farmers from engaging in the liquidation of productive assets which can have consequences for future production. Farmers with access to off-farm income, hence likely to be less liquidity constrained, are less likely to borrow from family and friends to cope, but are more likely to reduce consumption.

Farmers with access to radio, and hence more likely to be informed of potential changes in climatic conditions/risks and job opportunities in their vicinity and nearby communities are less likely to deplete/erode their productive assets nor promote migration, but are found to be adopters of food consumption reduction and borrowers of food, cash and other in-kind items from family and friends as coping strategies. This observation is in line with a report by [28].

After the analysis of the determinants of the choice of the coping strategies, interdependencies among the strategies were assessed based on the correlation matrix from the multivariate probit model. As shown in Table 3, a negative correlation was found between the liquidation of productive assets and a reduction in consumption. This implies that the former strategy may generally have a consumption smoothing effect in the study area. A positive correlation is found between reduction in consumption and migration, implying that households that opt for a reduction in consumption are likely to also encourage the migration of some members to the city in search of job. Migration and livelihood diversification are however found to be negatively correlated with borrowing, and this indicates that households that borrow from family and friends to cope with harvest failure are less likely to encourage migration of some members or to diversify their livelihood.

## 4.4 Determinants of the intensity of coping strategies

In this section, we present results on the determinants of the number of strategies adopted. From Table 4, it is found that the major determinants of the number of strategies adopted are farmer-to-farmer extension, the district dummy (location of the respondent), access to radio, access to off-farm income, access to government extension services, farmers' perception on the fertility status of their crop fields, number of family members abroad (outside the immediate community) and distance to the nearest market.

While variables like age, and value of farm implements also have significant effects on the number of strategies adopted, the effects are marginal. Access to relevant information, and training among other benefits through farmers' access to government extension services leads to the adoption of 0.19 less coping strategies. This in part could be attributed to the role such services play in reducing farmers vulnerability to shocks. Learning from colleague farmers through farmer-to-farmer extension however leads to the adoption of 0.66 additional coping strategies after the experience of harvest failure. Farmers with access to radio are also more likely to adopt 0.27 additional coping strategies, while farmers with access to off-farm income may adopt 0.24 less coping strategies. Having a positive perception about the fertility status of crop fields could lead to the adoption of 0.18 less coping strategies, while a km increase in the

**Table 4. Zero-truncated negative binomial regression estimates on intensity of coping strategies.**

| Explanatory Variables | Coefficients | Robust SE | Marginal effects | Robust SE |
|---|---|---|---|---|
| Age | -0.0021** | 0.0010 | -0.0071** | 0.0036 |
| Gender | 0.0174 | 0.0392 | 0.0583 | 0.1314 |
| Education of household head | 0.0036 | 0.0031 | 0.0121 | 0.0105 |
| Consumer/worker ratio | 0.0046 | 0.0368 | 0.0153 | 0.1234 |
| Family members abroad | -0.0127*** | 0.0044 | -0.0426*** | 0.0147 |
| Income from sec. liv. prod, /ME | 0.0001 | 0.0006 | 0.0003 | 0.0019 |
| Net value of liv. produced/ ME | 0.00002 | 0.00003 | 0.00006 | 0.0001 |
| Cropland / ME | 0.0164 | 0.0459 | 0.0001 | 0.0002 |
| Ownership of land | -0.0272 | 0.0468 | -0.0912 | 0.1567 |
| Produce crops on a subsist. basis | 0.0323 | 0.0301 | 0.1082 | 0.1009 |
| Value of farm implements | 0.0011**** | 0.0003 | 0.0037*** | 0.0011 |
| Fertility of crop field | -0.0547* | 0.0329 | -0.1833* | 0.1100 |
| Net crop income/ME | 0.00004 | 0.00006 | 0.0001 | 0.0002 |
| Severe yield loss also in 2013 | -0.0071 | 0.0305 | -0.0237 | 0.1023 |
| Access to credit | 0.0154 | 0.0351 | 0.0517 | 0.1177 |
| Access to gov. extension services | -0.0581* | 0.0349 | -0.1946* | 0.1165 |
| Distance to nearest market | -0.0095*** | 0.0032 | -0.0318*** | 0.0108 |
| Access to off-farm income | -0.0725** | 0.0301 | -0.2430** | 0.1004 |
| Group membership | 0.0109 | 0.0304 | 0.0365 | 0.1017 |
| Access to radio | 0.0792** | 0.0315 | 0.2656** | 0.1050 |
| Farmer-to-farmer extension | 0.1963*** | 0.0375 | 0.6580*** | 0.1248 |
| Bolgatanga | 0.1201*** | 0.0361 | 0.4026*** | 0.1198 |
| Constant | 1.1934*** | 0.1165 | | |
| /lnalpha | -17.336 | 0.0253 | AIC | 995.09 |
| alpha | 2.96e-08 | 7.47e-10 | BIC | 1083.90 |
| Log pseudolikelihood | -473.54 | | | |
| Wald chi2 (Prob > chi2) | 110.81*** | | | |
| Pseudo R$^2$ | 0.0144 | | | |
| Observations | 299 | | | |

Note:

***, **, and * represent 1%, 5% and 10% significance level

Source: Authors (based on output of zero-truncated negative binomial regression model)

distance to the nearest market could lead to the adoption of 0.03 less coping strategies. Having one more member outside the immediate community from whom the household may receive remittances leads to the adoption of 0.04 less coping strategies.

## 5. Conclusions

Strategies used by farmers to cope with harvest failure among other shocks have implications for future adaptation to such shocks. However, studies assessing farmers' vulnerability to shocks and their responses have been skewed toward documenting adaptation strategies and their determinants, with limited emphasis on the drivers of the choice and intensity of coping strategies farmers use to respond to those shocks. Using a survey data from 299 farm households in Upper East, a semi-arid region of Ghana, this study employed a multivariate probit model to analyze the determinants of the choice of coping strategies farmers use to respond to

harvest failure, and a zero-truncated negative binomial regression model to analyze the drivers of intensity of the coping strategies after harvest failure.

The study shows that most of the households relied mainly on liquidation of productive assets (livestock and other non-land assets), reduced consumption, borrowed from family and friends, diversified their livelihoods, and out-migration in search of jobs as coping strategies in response to harvest failure.

Although the liquidation of productive assets could assist in smoothening consumption and income, it could adversely affect production and investment decisions in the subsequent years. Besides the use of livestock as a liquid savings and its representation of wealth, it is found that the farmers who usually opt for this strategy earn no/low income from secondary livestock products (like eggs and draught services), live farther from market centers, have limited access to formal /informal credit, and have limited access to radio. The implementation of policy measures to promote farmers' engagement in the selling of secondary livestock products, and improve farmers access to credit, markets, and radio could contribute immensely towards reducing the likelihood for farm households to erode their productive assets in times of adversity. Having access to relevant production and livelihood related information through improved access to radio, access to credit, having a positive perception about fertility status of crop fields, and ownership of land were also instrumental in reducing the rate of out-migration to cities after harvest failure, which tended to minimize the over-burdening of women with farm work. Efforts to enhance farmers' access to vital production, market and job-related information via radio, ensure land tenure security, enhance farmers access to credit and increase the fertility of crop fields in the study area, could help to minimize out-migration after a harvest failure.

Beside these, the study found heterogeneous effects of the explanatory variables considered in the analysis on farmers' reduction of food consumption, borrowing from family and friends, and diversifying their livelihoods as coping mechanisms. Among the key observations are that the aged are less likely to reduce consumption or diversify their livelihoods due to their less mobile nature. The coping strategy adopted also depends on the net crop income per ME, the net value of livestock per ME, the value of farm implements held, previous/past experience of harvest failure and the orientation of crop production, with subsistence producers found to reduce food consumption as a coping strategy, but less likely to adopt livelihood diversification. These observed differences in effects should be taken into consideration in the drafting and implementation of measures to reduce farmers' vulnerability in the study area to harvest failure. The key determinants of the number of strategies adopted by farmers in the study area are farmer-to-farmer extension, the district dummy (location of the respondent), access to radio, access to off-farm income, access to government extension services, farmers' perception on the fertility status of their crop fields, number of family members abroad (outside the immediate community) and distance to the nearest market. Policy makers and stakeholders could make smallholder farmers less vulnerable to harvest failure by enhancing their access to radio, credit, off-farm income and market, promotion of farmer-to-farmer extension, implementing measure to improve the fertility of crop fields in the study area, and enhancing farmers' engagement in the production and selling of secondary livestock products.

While the study may be informative, it is not without limitations. Future studies could explore the possibility of a panel data analysis to explain the dynamics in farmer responses to harvest failure. In addition, it would be interesting to have analysis on coping strategies by farming households across countries and agro-ecological zones experiencing severe harvest failures over the years to ensure comparisons of country-specific and agro-ecological specific policies to enhance resilience.

## Supporting information

**S1 Appendix.**
(DOCX)

**S1 File.**
(DOCX)

**S1 Data.**
(DTA)

## Acknowledgments

Support offered by Drs. Justice A. Tambo, Vincent N. Kyere, and Messrs. Aaron Aduna, Baba Kunde, Samuel Ayaburi, and Matthew Sulemana is duly acknowledged and highly appreciated.

## Author Contributions

**Conceptualization:** David Boansi, Victor Owusu.

**Data curation:** David Boansi.

**Formal analysis:** David Boansi, Victor Owusu.

**Investigation:** David Boansi, Victor Owusu.

**Methodology:** David Boansi, Victor Owusu.

**Project administration:** David Boansi.

**Resources:** David Boansi.

**Supervision:** David Boansi, Victor Owusu.

**Validation:** David Boansi.

**Writing – original draft:** David Boansi, Victor Owusu.

**Writing – review & editing:** David Boansi, Victor Owusu, Enoch Kwame Tham-Agyekum, Camillus Abawiera Wongnaa, Joyceline Adom Frimpong, Kaderi Noagah Bukari.

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
