## [Decision Letter · Decision Letter 0]

1 Nov 2022

PONE-D-22-23073Responding to harvest failure: Understanding farmers coping strategies in the semi-arid Northern GhanaPLOS ONE

Dear Dr. Boansi,

Thank you for submitting your manuscript to PLOS ONE. After careful consideration, we feel that it has merit but does not fully meet PLOS ONE’s publication criteria as it currently stands. Therefore, we invite you to submit a revised version of the manuscript that addresses the points raised during the review process. Both reviewers have identified significant concerns with the methodology in the manuscript. These issues span study site selection, sampling, the choice of statistical methods and their application, and the missing qualitative data collection and analysis. All will need to be resolved, with subsequent changes to other sections of the manuscript needed in light of addressing the methodological issues.   Please submit your revised manuscript by Dec 16 2022 11:59PM. If you will need more time than this to complete your revisions, please reply to this message or contact the journal office at plosone@plos.org. Please include the following items when submitting your revised manuscript:A rebuttal letter that responds to each point raised by the academic editor and reviewer(s). You should upload this letter as a separate file labeled 'Response to Reviewers'.A marked-up copy of your manuscript that highlights changes made to the original version. You should upload this as a separate file labeled 'Revised Manuscript with Track Changes'.An unmarked version of your revised paper without tracked changes. You should upload this as a separate file labeled 'Manuscript'.

We look forward to receiving your revised manuscript.

Kind regards,

Claire Helen Quinn

Academic Editor

PLOS ONE

Journal Requirements:

“This study was funded by the German Federal Ministry of Education and Research (BMBF) through the West Africa Science Service Center for Climate Change and Adapted Land Use (WASCAL). Support offered by Drs. Justice A. Tambo, Vincent N. Kyere, and Messrs. Aaron Aduna, Baba Kunde, Samuel Ayaburi, and Matthew Sulemana is duly acknowledged and highly appreciated. ”

“This study was funded by the German Federal Ministry of Education and Research (BMBF) through the West Africa Science Service Center for Climate Change and Adapted Land Use (WASCAL).”

“This study was funded by the German Federal Ministry of Education and Research (BMBF) through the West Africa Science Service Center for Climate Change and Adapted Land Use (WASCAL).”

Additional Editor Comments:

Both reviewers have identified significant methodological issues with the manuscript, these will need to be resolved, along with the consequences for the results and discussion.

Reviewers' comments:

Reviewer's Responses to Questions

**Comments to the Author**

1. Is the manuscript technically sound, and do the data support the conclusions?

Reviewer #1: Yes

Reviewer #2: Yes

2. Has the statistical analysis been performed appropriately and rigorously? 

Reviewer #1: No

Reviewer #2: Yes

3. Have the authors made all data underlying the findings in their manuscript fully available?

Reviewer #1: Yes

Reviewer #2: Yes

4. Is the manuscript presented in an intelligible fashion and written in standard English?

Reviewer #1: Yes

Reviewer #2: Yes

5. Review Comments to the Author

Reviewer #1: This is a good study and the authors have significantly shown the determinants and extent of coping strategies adopted by the farmers. However, I have the following concerns about the methodology and conclusion.

The authors did not justify the use of poison regression in determining the intensity of coping strategies instead a Zero truncated poison regression. The data as presented in figures 4A and B showed that all the 300 farmers adopted one form of coping strategy or another meaning that there is no zero value, hence using a zero truncated poison regression should be a more robust tool rather than poison regression.

The sampling should be elaborated further to provide understanding and ease of replication. For instance, the list of farm households from the ministry of food and agriculture does it cover the 13 districts of Ghana, and how was the systematic selection implemented? Was it the 5th or 10th... occurrences etc that were selected?

The conclusion (from lines 561 – 563) that the “findings from the study thus suggest that asset-based theory may not necessarily hold since the farm households depended on the liquidation of livestock and farm implements as rapid coping responses to harvest failure” is not informed by the study. What do you mean by “asset-based theory”? this was not discussed earlier in the study. Assuming you are referring to "Asset-based welfare theory, in that case. your conclusion appears to be the opposite. The data presented showed that 94 percent of the farmers relied on assets as a coping strategy.

it will be informative if the researchers explained how they corrected for the potential effect of a similar experience in the preceding year instead of simply stating it (Ln. 347).

If the experience of harvest failure focused on experiences between the years 2005 to 2014 as stated in line 351, how do you relate the socioeconomic characteristics collected between February and May 2015 (Ln. 347) to an experience and response which occurred in 2007. For instance, a farmer might have experienced crop failure in 2007 and adopted a coping strategy the same year, however, acquired land in 2014, how do you justify that coping response adopted in 2007 or 2008 is determined by land ownership (Ln 471), the same goes for other socioeconomic variables.

Reviewer #2: 1. Authors trying to justify their study asserted that most studies concentrate on adaptation or adaptive strategies leaving out coping strategies. This might not be true as there a lot of studies that have looked at coping strategies in the area. Also, the variables measured by authors as coping strategies are the same measured in the studies, they are saying have concentrated on coping strategies. So it might be a matter of semantics. Therefore, authors need to revisit the problem statement and identify clearly what they are trying to do.

2. In the study area description, authors use statistics without making references to them (see line 326 and 327). These statistics are from other works and must be cited.

3. Methodology

The methodology has many flaws. First of all, in lines 273 and 274, another state that both qualitative and quantitative methods are used and yet, throughout the work, no qualitative method nor results presented. What qualitative data were collected? What qualitative method was used and how were the qualitative data analysed? The selection of the districts was skewed. All districts selected are in the central part of the region. With the exception of the Kasenas, the tribes occupying these districts virtually have the same cultural practices. However, the West (Bulsas) and the East (Kusasis and Mamprusis) are left out. Consideration of tribe/ethnicity is very important as culture affect the kind of adaptive strategies or coping strategies. Why did they concentrate on the middle? There should be a justification. Authors should note these East and West districts are the food basket of the region and any study of this kind should not leave them out. Also, the simple random and systematic selections are not described in detail. How was the simple random sampling done? How was the systematic Selection done? How many communities in each district was selected and why? Were these communities also selected by the simple random or systematic? How many farmers in each community were selected and why? All these questions need to be addressed in the methodology.

4. Results

#1. There are equations in the methods, but the results are not presented to reflect results from the equations. Most of the presentations can be done with cross-tabulation and that will show clear relationships than the results presented.

#2. Authors report that between 2005 and 2014, the year 2014 was the year with most crop failure. I turn to disagree with this because crop failure depends on many factors including climatic events such as floods and droughts. In 2007, the entire northern Ghana recorded the worst floods leading to the worse crop failure and food insecurity. This made the government of Ghana declared a state of emergency and called for external assistance to support northern Ghana. So to say that 2014 was worse is problematic. In fact, official documents show 2007 as the worse crop failure year. Authors should See many works such us Yiran and String 2016; Akudugu et al., 2012 and other related works. Also records from MOFA and NADMO can help. Their results might be as a result of memory loss, but authors needed to verify their response from official records and correlate with other works.

#3. The data was collected in 2014 and being published in 2022. That seems a bit old. This have changed and authors needed to at least go to the field to do rapid qualitative data collection to beef up. They are going to make recommendation which might not work since the data is not current.

#4. Also, some of the findings need some tweaking as there seem to be conflicting statements. For example, there is a negative association between age and reduction in consumption. This means as age increases, reduction in food consumption decreases. My knowledge of the area and the practice of the people seem opposite. This is because the youth have diverse ways of surviving while the aged due to low strength have limited options to survive.

#5. Also, Although the aged might not migrate, their children migrate and send them remittances. Remittances are very important coping or adaptive strategies and should be considered in the analysis to have a holistic view of the issues. Many of the youth are outside the region and send remittances to those left at home to buy inputs or food.

Conclusion

Conclusion is good but will need modification taken the above revisions into consideration.

6. PLOS authors have the option to publish the peer review history of their article (what does this mean?). If published, this will include your full peer review and any attached files.

Reviewer #1: **Yes: **CHUKWUMA UME

Reviewer #2: No

---

## [Author Response · Author response to Decision Letter 0]

19 Dec 2022

This has been attached/uploaded in the section dubbed "Attach Files"

---

## [Decision Letter · Decision Letter 1]

30 Mar 2023

Responding to harvest failure: Understanding farmers coping strategies in the semi-arid Northern Ghana

PONE-D-22-23073R1

Dear Dr. Boansi,

We’re pleased to inform you that your manuscript has been judged scientifically suitable for publication and will be formally accepted for publication once it meets all outstanding technical requirements.

Kind regards,

Zakari Ali, PhD.

Academic Editor

PLOS ONE

Additional Editor Comments (optional):

Reviewers' comments:

Reviewer's Responses to Questions

**Comments to the Author**

1. If the authors have adequately addressed your comments raised in a previous round of review and you feel that this manuscript is now acceptable for publication, you may indicate that here to bypass the “Comments to the Author” section, enter your conflict of interest statement in the “Confidential to Editor” section, and submit your "Accept" recommendation.

Reviewer #1: All comments have been addressed

Reviewer #2: All comments have been addressed

2. Is the manuscript technically sound, and do the data support the conclusions?

Reviewer #1: Yes

Reviewer #2: Yes

3. Has the statistical analysis been performed appropriately and rigorously? 

Reviewer #1: Yes

Reviewer #2: N/A

4. Have the authors made all data underlying the findings in their manuscript fully available?

Reviewer #1: (No Response)

Reviewer #2: Yes

5. Is the manuscript presented in an intelligible fashion and written in standard English?

Reviewer #1: Yes

Reviewer #2: Yes

6. Review Comments to the Author

Reviewer #1: Thank you for the great effort in improving the manuscript substantially. It is also interesting that you used the zero-truncated negative binomial regression model. The sampling and methodology are clearer to me now.

Reviewer #2: Revisit my earlier comments and work on them. My comments are very relevant. It will strengthen the paper.

7. PLOS authors have the option to publish the peer review history of their article (what does this mean?). If published, this will include your full peer review and any attached files.

Reviewer #1: **Yes: **CHUKWUMA UME

Reviewer #2: No

---

## [Editor Report · Acceptance letter]

6 Apr 2023

PONE-D-22-23073R1 

Responding to harvest failure: Understanding farmers coping strategies in the semi-arid Northern Ghana 

Dear Dr. Boansi:

I'm pleased to inform you that your manuscript has been deemed suitable for publication in PLOS ONE. Congratulations! Your manuscript is now with our production department. 

Kind regards, 

on behalf of

Dr. Zakari Ali 

Academic Editor

PLOS ONE